# The Life and Death of Freya the Walrus: Human and Wild Animal Interactions in the Anthropocene Era

**DOI:** 10.3390/ani13172788

**Published:** 2023-09-01

**Authors:** Abigail Levin, Sarah Vincent

**Affiliations:** 1Department of Philosophy, Niagara University, Lewiston, NY 14109, USA; 2Department of Philosophy, The State University of New York at Buffalo, Buffalo, NY 14260, USA; skvin@buffalo.edu

**Keywords:** animal ethics, climate ethics, sovereignty, capabilities approach, human–wildlife interactions, wildlife in urban areas, animals in cities

## Abstract

**Simple Summary:**

This paper addresses the killing of Freya the Walrus by the Norwegian fishing authorities in August 2022. Freya became famous for sunbathing on boats in the marina in the Oslo fjord, but she was soon euthanized in the name of public safety. Her death caused international outrage, and the aim of our paper is to demonstrate using philosophical argument why her death was unjust. We examine her plight through frameworks developed by animal ethicists involving co-sovereignty, capability, and individuality, concluding that any one of these frameworks, let alone several, would have led to a more just outcome for Freya. We argue that policy makers could put these insights into practice in a number of concrete ways going forward, as such incidents are likely to reoccur given the changes in migration patterns for animals in the Anthropocene era.

**Abstract:**

Freya the Walrus, who often climbed onto docked boats to sunbathe and frolic, was euthanized by the Norwegian Department of Fisheries in the Oslo fjord in August 2022, sparking international outrage and media attention. Since walruses are social animals, and since the Anthropocene era of climate change has displaced animals from their Arctic homes, forcing them to migrate, we can expect more human–animal interactions at such places as marinas, where Freya met her end. This paper asks and attempts to answer how we can make such interactions just going forward? In cases such as Freya’s, we need to reconcile three competing interests: the animal’s interest in living a flourishing life as best they can in a changing climate; the public’s interest in a safe and fulfilling wildlife encounter with an animal they have come to know intimately enough to name and follow devotedly on social media; and interests in maintaining private property. Examining these interests through the philosophical lenses of co-sovereignty, capability, and individuality, however, will yield more just results for animals in similar situations of conflict and co-existence with humans in urban spaces. We argue that, going forward, state resources should be expended to safeguard the public from marina access if safety is a genuine concern, while private money should be spent by marinas to enact safe animal removal with a no-kill policy.

## 1. Introduction

By the time of her death, Freya the Walrus—famous for her habit of nearing the water’s edge to climb onto docked boats at marinas throughout Europe in order to sunbathe and frolic—had an international fan base in person and online. She was euthanized by the Norwegian Directorate of Fisheries (hereinafter ‘Directorate’) at a marina in the Oslo fjord in August 2022, with an ostensible rationale of public safety. Her death sparked international outrage and media attention, suggesting a disconnect between the public’s priorities and those of the Norwegian government. In what follows, we ground and affirm the intuition behind the outrage—that her death was unjust—through philosophical argument.

Walruses are highly social mammals: “The walrus can be considered as the most gregarious pinniped for several reasons: group sizes can be enormous; walruses are gregarious throughout the year; and walruses are invariably in groups, whether hauled out on land or ice, resting in the water, traveling, or feeding” [1]. Since the loss of their ice floes in the Anthropocene has caused walrus migration away from their characteristic large groups, it is reasonable to think that Freya in particular found a facsimile for her herd in the socialization facilitated by her human fans. We can expect similar human–wild animal interactions in the future at such shared spaces as marinas. We hope to avoid such tragedies going forward, and so these arguments are offered in the spirit of providing conceptual resources for better policy and decision making in the future.

Marinas are liminal spaces of human–wild animal conflict and co-existence. Quite literally at the water’s edge, they are neither entirely land nor water; nor are they clearly public or private. Onlooking members of the public often have at least viewing access from the land, as they did in Freya’s case, while marina members have a straightforwardly private interest in their property and their access to it. Further, the public/private distinction is complicated because the state retains jurisdiction over the water and over public safety even if a particular marina might be a privately held enterprise. This liminality complicates how we think about justice in such cases, but we aim to clarify these ambiguities in what follows. After an overview of the end of Freya’s life and the role that anthropogenic climate change may have played in her migration in Section 2, we outline our key premises in Section 3. Section 4 and Section 5 situate our premises through reference to two central philosophical texts concerning human–wild animal interactions: Donaldson and Kymlicka’s *Zoopolis* and Nussbaum’s *Justice for Animals: Our Collective Responsibility*. However, more to the point, Section 4 invites consideration of Freya as a wild animal through our broadest lens of co-sovereignty. Section 5 shifts to consideration of Freya as a walrus in particular through the lens of capability. Section 6 narrows even more to Freya herself through the lens of individuality. We conclude our final sections with policy recommendations for similar cases going forward.

## 2. Freya’s Life and Death

### 2.1. A Timeline of Events

Freya was recorded traveling near the coasts of Sweden, Denmark, England, Germany, The Netherlands, and the Shetland Islands since 2019 [2]. Prior to arriving in Oslo, Freya was first named when she arrived on the Dutch coast in October 2021, resting on a submarine [3]. Her arrival marked the first time that a wild walrus had been spotted in The Netherlands in twenty-three years, as the species normally lives several hundred miles to the north in the Arctic [4]. She first appeared in Oslo on 17 July 2022 [3] and soon amassed an appreciable fan base of people who watched her climb onto boats to sunbathe. These pictures were often posted on social media, eventually giving rise to an international fan base [5].

On 20 July 2022, in its first public statement regarding the situation, the Directorate stated that it was exploring options to protect both Freya and the public, explicitly excluding culling as one of those options [6]. Though crowds and Freya’s celebrity continued to grow, in its second statement issued on 26 July 2022, the Directorate explained that, since walruses are a protected species in Norway, euthanasia was a last resort [7]. Norway, as a non-EU country that subsequently departs from some EU animal protection directives, has nevertheless ratified the Bern Convention (Appendix II), which lists walruses as a protected species. The statement of 26 July continued: “The conditions around her are calm, with few cases of close human encounters. Walruses do not normally pose a danger to humans as long as you keep your distance. However, when it is disturbed by humans and does not get the rest it needs, it may feel threatened and attack. Nearby people can provoke dangerous situations” [6]. Here, the concern appears to be public safety, with a noteworthy emphasis on human provocation being central to such public safety concerns materializing.

As time went on, the behavior of the onlookers indeed grew more varied, and though many onlookers just wanted to see Freya and take pictures of her, some others swam near her and threw things at her [8]. On 12 August, the Directorate issued what would be its third and final statement, this time stressing that euthanasia was now an option [8]—though the Directorate still explicitly stated that this was a “last resort” [5]. Less than two days later, late at night on 13 August, the Directorate shot her in the marina while she was on board a boat. Her body was chopped into pieces, with some samples taken for research and the rest dissolved into a large vat with lye [9]. It is vital to note that the euthanization was in spite of the 20 July and 26 July statements and the ratification of the Bern Convention; there was, in fact, no attempt to protect Freya—such as by relocating her—prior to euthanizing her.

When the decision to euthanize her was made, it was made in the name of public safety and only revealed to the public after the fact. The Directorate said: “The decision to euthanize the walrus was made based on an overall assessment of the continued threat to human safety… The public has disregarded the current recommendation to keep a clear distance to the walrus” [8], thus establishing a discourse whereby the Directorate’s agency was displaced onto the bad actors among the observers. Whatever the intention behind the statement, this discourse served to elide the under-discussed interest in private property protection (most especially, protection of the boats). Freya weighed approximately 1300 pounds and would often sink the boats she climbed upon [10], and it was not at all clear whether boat owners could claim the loss on their insurance, adding to their sense of perceived nuisance [10].

In the aftermath of the killing, when the public outrage reached its peak, the Directorate revealed the steps it had taken prior to euthanizing Freya, and this importantly included consultation with the Institute of Marine Research (hereinafter, IMR). Several options were canvassed, including anesthetizing her before attempting to relocate, deemed to be inadvisable due to the possibility of her drowning; moving her by putting a net under a boat, discarded because of the risk that she would become entangled in the net and panic; or building an open-top cage in which Freya could be relocated. The IMR was quoted as saying: “If [the cage option] had been successful, there would be little chance of the animal getting stuck under water or damaging the device” [11]. However, the IMR’s role was consultative only, and they did not make a final recommendation. After the killing, the Directorate said, with respect to having discarded all of the options, that “the extensive complexity of such an operation [i.e., any of the relocation strategies] made us conclude that this was not a viable option” [8]. They cited the “significant use of resources” involved in fashioning such a cage, though they did not give a projection for that expense nor justification for why it was deemed to be too much [11].

In the aftermath of Freya’s death, various social media pages [12] erupted in an outpouring of grief and anger, in part towards the Directorate and in part directed at the unruly spectators. Moreover, various experts shared these sentiments. Fern Wickson, a professor at the Arctic University of Norway said, after noting that “the risk was potential rather than demonstrated”, that the risk posed by Freya was no greater than those “we regularly tolerate in our society and daily lives. That the government chose to take Freya’s life rather than try to manage this potential risk through implementing more effective measures to manage the behavior of people was surprising and disappointing” [11]. Dr. Rune Aae, a biologist at the University of South-Eastern Norway, condemned the Directorate’s decision to euthanize her as “too hasty”. He said that fisheries staff were monitoring her with a patrol boat to ensure the public’s safety and that she would have “sooner or later gotten out of the Oslo fjord, which all previous experience has shown, so euthanasia was, in my view, completely unnecessary. What a shame!” [8]. Veterinarian Siri Martinsen, leader of the animal welfare organization Noah, said she was “shocked and very disappointed” to hear the news and that other measures should have been tried first; “We believe that those who fail to keep their distance from the animal should be fined, and I think that could have led to people following the guidelines. That has not been attempted” [13]. There were obviously other measures, such as fines, that would have incentivized the public to behave differently, protected Freya, and even potentially offset expenses associated with her protection or relocation. However, the Directorate did not try to implement any such measures, which strongly suggests that public safety was not actually the Directorate’s main interest.

The brevity of the relevant timeline—less than a month between Freya’s arrival in the fjord and her killing—suggests that the public was misled about the likelihood of euthanasia. Again, all three of the Directorate’s statements said that euthanasia was a last resort, and the walrus is a protected species, so the public would not have had much reason to fear this violent and extreme end. This blindsiding was likely contributive to the resulting outrage, to say nothing of the killing itself.

### 2.2. The Role of Anthropogenic Climate Change

The timeline of events above gives rise to the question of why Freya migrated to Oslo at all. There is widespread consensus in the scientific community that climate change has forced the migration of Arctic mammals [4,14,15]. Specifically, it is likely that boats mimic the ice floes of the Arctic that are now melting from climate change, hence her fondness for them. Further evidence for such conjecture is that Freya was not alone in her travels; two other walruses had earlier made similar voyages: Wally, who eventually returned to Iceland, after having had two pontoon boats built for him as safe spaces while on his journey [16], and Stena, who died after an injury caused by a fishing boat, despite an attempted rescue [17]. It is noteworthy to contrast these responses to Wally and Stena’s arrivals to the response to Freya’s; the UK went to great expense to build boats for Wally, and Finland attempted an elaborate rescue for Stena [17], but Norway summarily euthanized Freya.

While we cannot be certain that these migrations were caused by climate change, it is certainly very plausible, especially because experts attested that: “As climate change causes ice cover in the Arctic to melt, more walruses are hunting on land, creating competition for food that may explain why Freya has traveled so far” [10]. If the climate change hypothesis is correct, we must keep front and center in our minds that it is not climate change per se but anthropogenic climate change in particular that caused Freya’s migration, thus altering the calculus of how to balance what we characterize in what follows as the competing interests at play in this situation. If humans are to blame for her predicament, then surely what we owe her is different—and greater—than if she were simply a wild animal in need of help through no fault of ours.

Further adding to the need to weigh the situation carefully, it has been suggested that, granted the direction of her travel, she may have been consuming an invasive species of Pacific oyster [9], such that “her culinary tour could prove environmentally ‘helpful’…” [10]. This suggestion that Freya, albeit unintentionally, might have been performing an environmental service is an important one. Even if some of her behavior—climbing upon private boats to sunbathe and nap—could be considered a nuisance to the boat owners, her behaviors (if in part the result of anthropogenic climate change and if also having environmental benefit) cannot fairly be thought of as a mere nuisance. Here, conservation concerns might be relevant, granted the competition between walruses and oysters. One could argue that Freya’s actions were ecologically beneficial if she was consuming an invasive species.

This brief discussion of the facts, timeline, and issues at stake in this complex situation makes it clear that the decision to euthanize Freya was not straightforward. In what follows, we point a way toward a more just framework for negotiating situations of human–wild animal conflict and co-existence in the Anthropocene era.

## 3. Key Premises

We now set aside the question of whether we should treat Freya’s case as a matter of compassionate conservation since there was not a competing species or individual non-human animal in play in this scenario (aside from the oysters briefly discussed above). Given that this was primarily a case about a single animal and human interests, we believe that the Directorate needed to weigh the following four competing interests in adjudicating the proper response to Freya’s arrival in the fjord: Freya’s interest in living a flourishing life as best she could in the changing climate that we humans caused; the public’s interest in a safe wildlife encounter with her; (some of) the public’s interest in an emotionally fulfilling wildlife encounter with an animal they came to know quite intimately—enough to name her and follow her devotedly on social media; and the boat owners’ interests in maintaining their private property. As we have already seen to some extent above, it is unlikely that they weighed Freya’s or the public’s safety highly because they did not employ fines, barriers, animal welfare personnel, or other measures prior to quickly deciding to euthanize her, though there were no reports of her actually endangering humans. It is also unlikely that the Directorate weighed the public’s interest in an emotionally fulfilling relationship with her given the timeline, the deceptive nature of the Directorate’s statements, and the manner of her killing and the disposal of her body. All that remains, then, is the property interests in the boats, in which the Directorate must, by process of elimination, have been quite interested. For our purposes, we remain neutral on whether harm is suffered by the property itself or by its owners—but, in either case, we grant that property-based harms are possible.

So, how can we weigh these four competing interests in a more just and complete way? We outline three distinct philosophical lenses through which to view this question, each operating at a different level of specificity: the first, the lens of co-sovereignty, views Freya most generally as a wild animal (with us asking, “What do we owe to co-sovereign communities?”); the second, the lens of capability, views Freya as a member of a particular species—that is, as a walrus (with us asking, “What are her species-specific goals for a flourishing life, and how can we aid them?”); and the third lens of individuality views Freya as the individual we named and with whom we interacted (with us asking, “What special obligations do we owe to her in light of her own particular plan of life?”). Through this examination, we come to see that had any of these levels of analysis, let alone several in combination, been taken on board by the Directorate in its reasoning, it would have yielded a better outcome for Freya. Indeed, it is a strength of our argument’s structure that one need not accept all of our three lenses of analysis as plausible or compelling; even one is sufficient to have warranted a different interaction with her.

Before beginning our discussion of these three lenses in the sections that follow, we would like to note three more general background premises that inform our argument throughout. Nussbaum recognizes at the outset of *Justice for Animals* what we take as a fundamental starting point in thinking about human–wild animal interactions in general—and Freya in particular: that whatever grounds our human right to use the planet for survival likewise grounds the same right for animals.

…all animals, both human and non-human, live on this fragile planet, on which we depend for everything that matters. We didn’t choose to be here. We found ourselves here. We humans think that because we found ourselves here this gives us the right to use the planet to sustain ourselves and to take parts of it as our property. But we deny other animals the same right, although their situation is exactly the same… Typically, no argument at all is offered for that denial[18] (p. 2).

This insight about our shared human–animal predicament, and humans’ asymmetrical response to it, begins to explain the intuition that many of us had that Freya’s death was unjust. Freya was trying to make the best of her existential predicament—migrating after having been thrown into existence at this precarious moment of the Anthropocene—just as we are. She is thus, from Nussbaum’s perspective, to be seen as on equal footing with us humans in the competition for scarce resources. In our view, we as humans thus had the obligation, which we discharged in an at best cursory manner, to consider her predicament along with ours in deciding whether and how to intervene in her life and also in deciding how to understand her use of the resource in question (i.e., the boats).

To the extent, then, that we view Freya as encroaching on ‘our’ human space and resources, we see the matter ahistorically, disregarding our incursions into ‘wild’ space, most especially over the past sixty or so years, and through a colonialist lens, through which such human incursions are seen as justified or even invisible. However, this is clearly false. In this vein, Donaldson and Kymlicka note that the human population has more than doubled since 1960 while wild animal populations have dropped by a third over the same time period due in large part to human incursion into formerly wild space [19] (pp. 192–193).

Combining Nussbaum’s and Donaldson and Kymlicka’s ideas, we come to our second key background premise: a rejection of the naive idea that some spaces are clearly wild, in the sense of untouched by human beings, while others are ‘civilized’, ‘built’, or ‘settled’ by humans. Given that anthropogenic climate change alone—to say nothing of other human incursions—leaves no corner of the Earth untouched by humans, it follows that:

…there is no “wild” in the intended sense: animals can be free of human incursions only when humans act benignly to protect their spaces. The use of the term “wild” today is usually a pretense that we are not there, and thus functions as a way of evading our responsibility to make our (ubiquitous) domination benign rather than hideous. Getting clear about our omnipresence makes it urgent to think about what relationships are open to us other than those of brute domination[20] (p. 87).

In place of this disingenuous and even arguably genocidal notion of ‘wild’, Wichert and Nussbaum propose the following: “Let ‘wild’, then, simply mean “not systematically bred to be interactive with and companionate with humans. This is a definition that…applies to many species in today’s world, and does not involve us in hypocritical denial of human domination” [20] (p. 89). We accept this renovated definition of ‘wild’.

Our final background premise is that anthropogenic climate change generates a moral responsibility for it and a duty to repair it on behalf of the humans who caused it with respect to (at least) those sentient beings who are victims of it. We do not argue for this premise here as a full defense of it would take us too far afield, and we acknowledge the complications of conflating ‘those who caused climate change’ with ‘those who (at least collectively) contribute to it’ and of setting aside the non-sentient world in our present discussion. However, we believe that Donaldson and Kymlicka and Nussbaum offer quite a bit of argument for it in what follows.

These three premises—first, the idea that animals have the same right to use the planet for survival as humans; second, that there is no clean division between ‘wild’ and ‘civilized’ space; and third, that we humans have moral responsibility for anthropogenic climate change—would have necessitated a more just outcome in Freya’s case had they been weighed by the Directorate. These premises also undergird the three philosophical lenses through which to view situations of human–wild animal conflict and co-existence, as we will see as we now discuss each in turn.

Before turning to these discussions, we want to add a note of clarification about our use of the term ‘we’ in what follows, specifically in instances where claims are made about obligations to act. We (as co-authors) acknowledge that the obligations born by a wider ‘we’ could plausibly vary based on environment (e.g., urban or rural), socioeconomic class, and community affiliation or sovereignty status (e.g., Indigenous). For our purposes, the obligated ‘we’ here is constrained to liberal, democratic societies roughly similar to Norway. Much of what follows is inspired by the writings of Donaldson, Kymlicka, and Nussbaum—precisely because we think their approaches are relevant to this narrowed audience.

## 4. Freya and Co-Sovereignty

Donaldson and Kymlicka propose to treat wild animal communities as sovereign nations—that is, with the ability and right to govern themselves—on equal footing with our own human sovereign communities. This is the first level of generality of our analysis; co-sovereignty, also known as overlapping sovereignty, applies to any wild animal species, irrespective of that species’ particular form of life. Viewing human–wild animal interactions through a sovereignty lens allows us to view “the rights of wild animals in terms of fair interaction between communities” [19] (p. 167). Of course, methods of self-governance take different forms between different species, but it is important to remember that they do as well between different human nations. We can see how sovereignty is an apt lens through which to view human–wild animal conflict and co-existence in spite of the surface differences in styles of governance when we ask what grounds the notion of sovereignty altogether. In doing so, we find that it runs much deeper than particular forms of life and styles of governance. Donaldson and Kymlicka astutely note that sovereignty is linked to a community’s interests in maintaining their distinctive form of social organization on their own territory and to a community’s interests in protecting against particular threats, especially external rule and territorial incursion [19] (p. 180).

Viewing it in this way, it is clear that wild animal communities have interests in maintaining their social organization on their territory and face threats—in particular, the threats of human incursion on that territory. It is also clear that they already govern themselves accordingly to further their interests and mitigate against threats. Human recognition of their self-governance as a form of sovereignty allows us to treat these communities as respectfully as we would treat sovereign human communities.

However, all of this comes with an important caveat—that these communities cannot be understood as occupying geographically discrete spaces, in contrast to the norm for human sovereign nations, since: “Nature did not assign discrete territory to different species…Sovereignty, therefore, if it is to mean anything in practice, cannot be tied to a picture of neatly divided communities and territories” [19] (pp. 187–188). This observation is particularly apt for Freya’s case, where she was forced to migrate large distances into overlapping territories. The Directorate treated her movement as though it were an incursion on to clearly human space; but, through a co-sovereignty lens, it was anything but. This insight coheres with our second foundational premise, set out in Section 3: that there are no clear geographical distinctions to be made between ‘wild’ and ‘non-wild’ spaces.

Donaldson and Kymlicka note that keeping these fuzzy boundaries between humans and wild animal species at the forefront of the mind is often more difficult for humans when they are dealing with a water animal since we are so much more land focused, given our typical habitats [19] (p. 188), and this was clearly true in Freya’s case. Donaldson and Kymlicka continue in this vein:

Given that sovereign animal communities exist within, and in parallel with, human communities, all territory is, in a sense, border territory…[I]n regions of overlapping sovereignty (migration routes, mobility corridors, shared ecosystems, etc.), human communities are not free to pursue their interests in ways that ignore the interests of wild animal communities who are co-sovereign there[19] (p. 197).

Thinking of the marina as border territory between land and sea, and as having become part of a migration route for walruses due to anthropogenic climate change, we can see Freya’s situation much more clearly—and see our obligations much more clearly—when we look at her arrival and residence in the Oslo fjord as a case of overlapping sovereignty in a border territory. Once we see this, we see that we as humans should not have neglected our co-sovereignty as if Freya were a mere nuisance encroaching on ‘our’ territory. However, that is exactly what the Directorate did in euthanizing Freya. Once we extend sovereignty beyond human nations, Freya’s behavior is no longer an encroachment; through a co-sovereignty lens, we see she had a right to co-exist with us at the marina.

Further, the lens of co-sovereignty would have provided us with an obligation to aid Freya, likely by moving her from the marina in a craft fashioned for the purpose without killing her, as the IMR proposed [11]. Donaldson and Kymlicka write:

We owe duties of justice to wild animals. In general, respecting their sovereignty means we should be very cautious about undertaking interventions in nature. But respect for sovereignty is consistent with undertaking many individual and time-limited or scale-limited acts of assistance…When we can help animals without usurping their autonomy or causing greater harms, we should be moved by the plight of suffering individuals[19] (p. 187).

We had just this opportunity to aid with the option to build a craft to re-home Freya. This possibility was considered and dismissed by the Directorate as too costly [11]; but when viewed through a sovereignty lens, cost ceases to be a deal-breaking issue. After all, we regularly employ vast governmental and military aid, requiring very complex logistics and huge expense, to disaster situations in foreign or remote sovereign human nations. In Freya’s case, at least at the most immediate level, the aid in question was building a custom, open-top cage. A cost estimate for such a cage was not released by the Directorate, as far as we can determine, but it is reasonable to assume that it would not have exceeded tens of thousands of dollars—a trivial amount for a wealthy country such as Norway.

So, a sovereignty lens imposes a duty to assist (where such assistance does not violate autonomy or self-determination [19] (p. 178)); and, here, a safety-preserving crate for moving Freya would have been likely effective and in line with that caveat. However, further, a sovereignty lens allows us to better weigh some of the competing interests in Freya’s case (as outlined in the previous section). Donaldson and Kymlicka write:

We should note that these risks are not all in one direction. Wild animals can pose a threat to human activity (e.g., road collisions with deer or moose…), or to public health (e.g., animal viruses), or indeed from attack (e.g., from grizzly bears…). Such risks are inevitable so long as both humans and wild animals continue to share the planet. And so a crucial task of a theory of animal rights is to determine the appropriate principles for regulating these risks[19] (p. 197).

The very fact that Donaldson and Kymlicka look at risks as bilateral—to and from humans and wild animals—rather than unilateral—from wild animals to humans only—is, in itself, quite radical and should be seen as a stark contrast to the ways in which the risks were assessed by the Directorate.

However, Donaldson and Kymlicka’s account goes quite a bit further, and, in so doing, is able to illuminate the issues underlying the injustice in Freya’s case. Most importantly, we cannot continue with our idea that risk bearing ought to be unilateral—we accept zero risk from wild animals while they accept untold amounts of risk from us:

We tend to look at any risk posed by wild animals to humans as unacceptable. However, considering the enormous risks we pose to animals, it is unreasonable to expect zero risk in the opposite direction. Instead, we should accept a certain level of risk from the presence of wild animals in areas of overlapping sovereignty… human communities do not have the right to eliminate the general risks posed by the presence of wild animals in overlapping zones… We cannot demand that these animals be culled in order to reduce our risk. In other words, we cannot demand zero risk for ourselves, at the same time that human societies impose extraordinary risks on wild animal communities[19] (pp. 202–203).

Putting the problem even more bluntly, Donaldson and Kymlicka write: “In general, the risks imposed on us by wild animals pale in comparison with the risks we impose on them. This incredible disparity of risks should ring alarm bells for our sense of justice” [19] (p. 201). The sovereignty approach insists that we are in a reciprocal relationship with other species, and the passage above insists that we stop denying that and stop acting *as if* we are entitled to insist on zero risk from the animals with whom we share the Earth.

If we had taken the idea of our obligation to share risk seriously, we would not have euthanized Freya. Granted, her proximity to humans may have posed some risk, but if we had taken account of the fact that we were in shared space to begin with, and, therefore, that we were not entitled to zero risk, and, further, that we were likely the cause—as the agents of anthropogenic climate change—necessitating her migration, we would have weighed the risks to us—whether to our physical person or to our property—less heavily in our decision making.

The Directorate euthanized Freya in the name of her risk to public safety, but it was conceded that that physical risk was understudied [11] and minimal, at least under typical conditions. The very crucial issue of our risk to her from anthropogenic climate change was utterly disregarded in the calculus. So, too, were the pleasures of both humans and animals overlooked—Freya’s pleasure in socializing and sunbathing, and the onlookers’ pleasure in watching her. Recall that these pleasures constitute two of the four competing interests at play in this situation: Freya’s attempt to flourish in the Anthropocene and humans’ interest in having a fulfilling encounter with her. The real risk, it seems, was the risk to the boat owners’ property; and that was given very outsize weight when viewed through a co-sovereignty lens.

Instead—and this is vital in thinking about such cases going forward—Donaldson and Kymlicka propose that a sovereignty lens treats the issue of risk distribution as a matter of justice, as between co-sovereign communities, and, thus, that the imposition of risk on one community by another must meet a number of conditions, including:(a)The imposed risks must be genuinely necessary to achieve some legitimate interest and are proportional to that benefit;(b)Both the risks and the attendant benefits must be equitably shared overall—the people who suffer risk in one context must benefit from risk in other contexts;(c)Society must compensate, where possible, the victims of inadvertent harm [19] (p. 198).

These criteria offer concrete metrics by which to assess whether risks imposed on sovereign communities are just and allow us to see quite clearly how Freya’s case breached them. As against criterion (a), the risk to Freya was no less than death, weighed against the relatively trivial human interest in private property, whatever risk to human physical safety Freya may or may not have posed, euthanasia in advance of attempting fines, barriers, etc., to protect human safety was certainly disproportional to that interest. With respect to (b) and (c) in Freya’s case, it is obvious that we failed even more egregiously. With respect to (b), the ‘group’—here, wild animals caught in encounters with humans in liminal spaces—is continually ‘the victim of imposed risk’, rather than the risk being shared equitably between the two groups. While it might seem, *prima facie*, that humans are at equal risk in human–wild animal encounters, we have many more resources to protect ourselves; in this case, through behavior modification and physical barriers. The wild animal, conversely, remains at the mercy of human negligence or maliciousness and vulnerable to the state’s resources and power. With respect to (c), obviously, a dead animal cannot be compensated for harm.

Thus far, in Section 4, we have examined Freya as a wild animal in general, and, as such, as a member of a sovereign community. In this capacity, she is a proper recipient of a duty to aid and of a recognition that humans and wild animals pose mutual risks to one another, and any mitigations of these risks have to be carefully and justly calibrated.

## 5. Freya and Capability

Where Donaldson and Kymlicka’s sovereignty approach centers on what we as communities owe to each other, Nussbaum’s Capabilities Approach (hereinafter, CA) provides a fruitful lens through which to view Freya’s plight as a walrus, that is, as a member of a particular species with a particular way of life and goals for fulfilling that way of life. Nussbaum begins her recent book *Justice for Animals: Our Collective Responsibility* by providing a general account of injustice, equally applicable to non-human or human victims. Injustice intuitively involves “the idea that someone is striving to get something reasonably significant, and has been blocked by someone else—wrongfully, whether by malice or by negligence” [18] (p. 1). This aptly encompasses Freya’s plight—the something reasonably significant that Freya was striving for was twofold: a home and socialization. She was thwarted from achieving these ends by the Directorate. We are inclined to argue that negligence is the relevant harm here rather than malice since the Directorate did not take the time to make a more careful decision regarding re-homing her. She was, again, a victim of negligence by human society at large, as we consumed the vast amounts of fossil fuels that led to the anthropogenic climate change that caused her to seek a new home at the outset.

Nussbaum’s idea of striving towards something significant points us toward her central idea of capabilities. As she puts it, “Animals, like humans, have, each of them, a form of life that involves a set of important goals toward which they strive” [18] (p. 96). Once an animal has achieved the goals for which they strove, they can be said to be in a state of flourishing: “Essentially, then, the CA is about giving striving creatures a chance to flourish. For the capability theorist, a chance to flourish means not just avoiding pain, but a list of positive opportunities…” [18] (p. 81). In this framework, as she puts it:

Capabilities are core entitlements, closely comparable to a list of fundamental rights… it emphasizes material empowerment more than do many rights-based approaches, however, it leaves spaces for individual freedom: someone who has all the basic opportunities or entitlements spelled out in the list of capabilities is not required to act on them. The choice to act is left up to them[18] (p. 80).

Nussbaum is arguing for a robust sense of the good life where what matters is not just the absence of pain but a fully articulated and realized sense of the pleasures that life may afford a particular kind of animal. As humans who share the Earth with animals, we are uniquely able to provide such resources to enable each animal to live a flourishing life, and Nussbaum argues that it is our collective responsibility to do just that.

The capabilities of animals of course vary from species to species, and our human knowledge of them will grow as research advances. That said:

…there are likely to be capabilities that are particularly fertile, promoting good life across the board, and capability failures that are especially damaging. For all animals, subjection to arbitrary human violence is a corrosive disadvantage… Another corrosive disadvantage across the board is environmental pollution, which causes lethal conditions, whether of air or water, for many species and depletes their habitats. So the opposite of these ills—bans on cruel practices and a dedication to environmental cleanup—will prove fertile, enhancing capabilities across the board for many animals[18] (p. 105).

This discussion of course encompasses the specific violence of shooting Freya, but also the background violence of destroying her habitat through fossil fuel consumption as both involved thwarting her capabilities and subsequently gave rise to injustice.

If Freya’s migration was indeed an effort to flourish in the Anthropocene, then, according to Nussbaum, we ought to have joined Freya in her effort to accomplish her aims—or at the very least have not actively thwarted her. According to Nussbaum’s account, we ought not to have euthanized Freya, as such action was an active thwarting, and we, further, have the positive obligation to restore her habitat so that migrations such as hers are no longer necessary. In an intermediate position, we had, as we saw in the preceding section on sovereignty, a duty to aid in the particular case, which could have been discharged through fashioning a crate in which to move her.

Regarding such a duty to aid, we noted an important difference between humans and other animals in the last section: namely, that human sovereign communities tend to be more obviously distinct from one another. (This is of course not always the case, as exemplified in tensions between Indigenous persons and successful colonizers.) Similarly, with respect to capability, human goals tend to be more easily recognized owing to shared language and to similarities in embodiment that facilitate meaningful gestures. As the lines dividing sovereign communities may blur when we move to thinking about non-humans, so too can their goals seem obscure to us in some cases. However, Freya had a straightforward purpose in coming to the docks—to achieve goals significant to her: opportunities to socialize, to sunbathe on an ersatz ice floe, and to eat delectable and invasive oysters.

Still, even when we can be sure of another’s goals, we often do not realize what Nussbaum calls the ‘capabilities list’—an accounting of species-specific goals—in the case of humans, let alone non-human animals. Obvious shortfalls occur disproportionately in different countries, and for different genders, races, abilities, and so on. Consider, for example, the difficulties around implementing successful accommodations for persons with disabilities or persons who are neurodivergent even when we recognize goals related to increased freedom of movement and effective learning environments. How, then, are we to turn our collective political will toward animals? Nussbaum notes that when we pay attention to how an animal lives—observing its teleological strivings—a sense of wonder awakens in us:

Wonder is connected to our perception of striving: we see that these creatures have a purpose, that the world is meaningful to them in some ways we don’t fully understand, and we are curious about that: What is the world for them? What are they trying to get? We interpret the movement as meaningful, and that leads us to imagine a sentient life within”[18] (p. 11).

Freya’s situation allowed us, her human onlookers and fans, a sense of wonder that engaged us with her project and motivated us toward assisting with her plight. Even more pointedly, when our wonder is aroused, and we then witness the object of our wonder’s strivings thwarted, “…that might lead to an ethically directed compassion when the animal’s striving is wrongfully thwarted, and a forward-looking outrage that says: ‘This is unacceptable. It must not happen again’” [18] (p. 9). Freya’s case, we maintain, awakened these very emotions, and that is why her death caused such a strong sense of outrage.

However, we can turn that outrage into a forward-thinking attempt to make the world better going forward. As Nussbaum puts it, once these emotions are aroused, all we need to do is turn our attention into thinking toward a positive solution: “In general: whenever we think we have to inflict hardship on animals in order to preserve a healthy human community, we ought to step back, asking how we got into that bad situation, and what we might do to produce a future world in which that grim choice does not arise” [18] (p. 192). Future-directed thinking is important here; we mishandled Freya’s case, but the point is that we can do better going forward: “I think we can deliberate well, and create a feasible multispecies world. The only question is: Will we?” [18] (p. 192). We hope that this paper, insofar as it offers forward-looking recommendations for how to weigh the competing interests at stake in these cases of human–wild animal conflict and co-existence, thereby offers resources whereby it might not happen again.

Returning now to our original issue of determining the best way to adjudicate between the four competing interests in Freya’s case, how might the CA be of use? It may not be as simple as determining what it would take for Freya, or another walrus, to flourish and providing her with those material resources that would enable her to do so. A cynical response to this sort of prescription might emphasize the points of conflict over some material resources, most especially the boats. In a chapter on what Nussbaum calls “tragic conflicts” between humans and animals, she offers this clarifying advice:

The first step in thinking well about such cases is to analyze the conflict clearly. Do both sides really involve pushing people below the threshold of a genuinely major capability? Not all interests are of equal weight. Humans, for example, can adapt to a smaller space more easily than can most large animals, as the success of cities has taught us; so we should not think that when humans are asked to do that in order to support the flourishing of an animal group, this is necessarily tragic. Nor do human financial interests by themselves generate tragedy[18] (pp. 189–190).

Freya’s situation failed this test. By all accounts, the physical risk Freya posed was low [11], so the human interest appears, by process of elimination, to have been financial—in the property of the marina members’. This does not rise to the level of a tragic conflict by Nussbaum’s definition, and, obviously, the decision to euthanize pushed Freya “below the threshold of a genuinely major capability”, which can now—when seen through the lens of the CA—be understood as clearly unjust.

Put simply, other options should have been considered in light of her capabilities as a plausibly sapient and clearly sentient mammal. Nussbaum sets out three ways that humans may meet our ethical burden to other animals, the first and third of which are relevant for our discussion:

First, and most urgent, humans must end human practices that directly violate wild animal life, health, and bodily integrity… Far more difficult is the third: humans must protect wild animal habitats from damages due to climate change and other environmental factors that likely have a human origin[18] (pp. 234–235).

The third step of attempting to make her Arctic habitat more hospitable in order to negate the need for migration altogether is admittedly difficult. However, in this case, our responsibility could have been easily discharged by taking the first route: by simply relocating Freya rather than euthanizing her, since euthanasia, by definition, violates life.

In Section 5, we have considered Freya as a member of a particular wild animal species striving to flourish according to her particular walrus-specific goals. As such, we owed duties not to thwart her aims and, moreover, to actively aid her in facilitating her flourishing, as per her species-specific capabilities.

## 6. Freya and Individuality

As useful as the previous two frameworks are for honing in on the proper characterization of our ethical failure in Freya’s case—we failed to respect her as a member of a sovereign community and we failed to facilitate the flourishing of her species-typical capabilities—they are not the full story. Indeed, what makes Freya’s particular story so compelling and the outrage at her death so palpable is that she was an individual—not merely a wild animal nor merely a walrus. Her highly idiosyncratic behaviors of climbing onto boats to sunbathe and nap and her extreme sociality were features of her as an individual, and they are what drew us to her. It is not perniciously anthropomorphizing to say that Freya had a personality, which she expressed through these behaviors. Consider that not every walrus is equally sociable, charismatic, or adventurous.

Both Donaldson and Kymlicka, as well as Nussbaum, allow for such further individual consideration of our ethical responsibilities to wild animals, and it is important to examine this individual dimension of Freya’s case in order to obtain a fuller sense of the ways in which her death was a moral failure on humanity’s part, as exemplified by the actions undertaken by the Directorate. In *Zoopolis*, Donaldson and Kymlicka consider several cases where individual humans formed relationships with particular wild animals and, consequently, are characterized as either owing special duties of care to these animals or as simply responding compassionately to their plight. In one case they discuss, a woman decides to feed deer who have gathered in her yard for some time, having first weighed all the ethical and environmental considerations of doing so, and decides in the end to help the deer simply because they are suffering and she is in a position to help them. They call this a response of “simple compassion” [19] (p. 183). In other words, a full theoretical consideration of all our ethical duties may be superseded by the fact of the development of an individual relationship. Feelings of compassion arise spontaneously rather than purely rationally in such cases. Characterizing the Directorate’s response as lacking in compassion is also an apt way of understanding it.

Donaldson and Kymlicka also discuss our individual relationships with wild animals as giving rise not solely to a compassionate response, but, in more traditionally philosophical terms, as giving rise to a duty of care—a duty that we otherwise may not have had. Discussing a woman who developed a relationship with some nearby beavers, they describe this duty of care as emerging out of a relationship she formed with the beavers and in response to the benefits conferred on her via those relationships [19] (p. 184). Freya provided many emotional benefits to the humans who enjoyed watching her, so a duty of care may be seen as having reciprocally arisen from these benefits. So, both compassion and a duty of care may give rise to a duty to aid particular individual wild animals once our interactions with them have given rise to considering them as individuals. We can consider compassion as an emotional, or perhaps pre-rational, response that arises spontaneously from the feelings evoked by the relationship, while we find that a duty of care arises once we rationally weigh what we owe the particular animals with whom we have formed the relationship and from whom we have derived benefits that justice demands we reciprocate.

Viewing particular wild animals as individuals, and recognizing duties that arise from our unique relationships with some of them, is compatible with either a sovereignty or a capabilities approach but not co-extensive with either. We can view Freya’s use of the marinas she visited and the boats on which she lounged as her attempt to express interests that were, at least somewhat, particular to her as an individual, not as a member of a species or a community. After all, not all walruses migrate as she did and make use of liminal spaces such as marinas. When the Directorate euthanized her, we—collectively as humans responsible for anthropogenic climate change—violated the ethical imperative to consider Freya as an individual with bodily autonomy in the most profound way.

Though Nussbaum’s Capabilities Approach focuses on species-typical capabilities, she is careful to note that this focus does not preclude obligations to promote, and to not thwart, behavior that is outside of the species-typical norm:

So far, I’ve spoken of a list [of capabilities] for each species of animal. But for animals, as for humans, each individual creature should be treated as an end. And animals are individuals not just numerically (each one matters), but also qualitatively: each species member is subtly different, one from every other[18] (p. 105).

As we noted previously, Freya’s highly idiosyncratic behaviors of sunbathing on boats and of seemingly relishing human interaction are key to understanding why many of us were fascinated with her and why that fascination led to us forming a relationship of sorts with her. That relationship, once formed, then gave rise, we argue, to positive obligations which the Directorate—acting as a state agency ostensibly on our behalf—utterly failed to meet when they made the decision to euthanize her. This individually focused approach could have wider implications as Freya is not the only high-profile case of an individual, named animal whose personality, and untimely demise, has attracted human interest (e.g., Cecil the lion, Marius the giraffe, Knut the polar bear, Harambe the gorilla).

That said, it is important to note that while these particularly charismatic animals, and the horrific occasions of their deaths, might elicit outsize responses and trigger more ethical outrage from the public than other killings of less charismatic animals, we should not be misled into thinking that we only have ethical obligations to animals who are charismatic and to whom we are emotionally attached. The strength of this paper’s argumentative structure is that one need not accept any special obligations that follow from such attachment and may instead take the more objective routes offered by Nussbaum or Donaldson and Kymlicka.

## 7. Policy Recommendations

Viewing wild animals as members of sovereign communities allows us to properly understand our duties to aid them; the Capabilities Approach allows us to view wild animals as having species-specific goals towards which they strive in order to flourish, and, in so doing so, provides us with a sound account of injustice when those goals are thwarted; and philosophical considerations of wild animals as individuals allow us to further contextualize the emotional upset in Freya’s specific case.

Viewing Freya’s situation through these three philosophical lenses allows us to more justly weigh the four competing interests in this case: again, Freya’s right to flourish; human interests in safety; human interests in forming a relationship with or at least having an experience with Freya; and human interests in private property. What these lenses show is that the human interest in private property should have been given the *least* weight, though, in Freya’s case, it was likely given the most. The rhetoric about protecting human safety, as we have shown [11], was overblown at best and disingenuous at worst. However, of course, the interests in property and human safety are worth something in our considerations, and we suggest that they could have been addressed in ways that did not require sacrificing the more important interests at stake, most especially Freya’s wellbeing.

How, then, might we arrive upon a more just and safe outcome in future situations of human interactions with wild animals in liminal spaces? We, as philosophers, are not in a position to be experts on policy making; the onus is on the state, private marinas, and/or insurance companies to come up with policies in advance given the increasing likelihood of situations such as Freya’s. A full-scale consideration of this exceeds the scope of this paper; we can nevertheless begin to sketch some policy recommendations here. In order to address the less important considerations—public safety (which is less important because it can be safeguarded by any number of simple interventions) and private property—we suggest that, going forward, state resources should be expended to safeguard the public at marinas if safety is a genuine concern, such as by installing plexiglass walls, for example, so the public can safely view wild animals, while private money should be spent by marinas to ensure safe animal removal with a no-kill policy. These latter costs should be paid in advance through a fund collected through member dues, analogously to money set aside for contingencies in a condominium Homeowners’ Association, so that we are not left in the future in the disorganized crisis in which the Norwegian authorities found themselves. Such an arrangement would also help navigate the confusion between the public and private interests at play in these types of cases.

With respect to the most important interest—Freya’s wellbeing—it is clear that the duty we owe is no less than the duty to curb anthropogenic climate change to protect against even worse habitat destruction in the future. Failing that, or in the interim, we need to accept Donaldson and Kymlicka’s key point that we are always in shared spaces with wild animals, that the risk we tolerate from them should not be set at zero, and that the risks we impose on them are quite large. If we had kept this in view during Freya’s migration, the annoyances of and potential financial losses to the boat owners could have been tolerated as a reciprocal risk we owe to a member of a sovereign nation. Insurance companies can also help us bear some of this risk by beginning to write policies that cover private property damage from wild animals.

One worry here is that certain agents, or collectives of agents such as insurance companies, may be resistant to policy revisions that do not have an apparent financial upside. However, at least some of our suggestions can be implemented in ways that can keep the profit margin in mind. For example, if a developer undertakes construction of a marina and a policy exists that requires some of that space to be set aside for wild animal use, this is beneficial to all parties. Such a policy could curtail damage to property, discourage more risky interactions with wild animals, and incentivize visitors to enjoy a still intimate visit with wildlife—but a safely distanced one. For example, a small admission fee might be charged to spectators by the marina to offset expenses incurred in designating safe(r) areas for human–wildlife interactions. Taking seriously the needs of wild animals in liminal spaces stands to benefit not only them but also us. While it may not be a part of the mandate of administrative agencies (such as the Directorate) at present to consider facilitating healthy human–wild animal interactions, going forward, it needs to be. The Anthropocene requires no less of us.

Moreover, regarding the Directorate in particular, as a state entity, they work for the people, who clearly voiced their interest in such encounters with Freya. Additionally, in the Anthropocene, as migration patterns change, enabling more human–wild animal encounters, mandates of state entities such as fisheries directorates need to change to accommodate relevantly similar phenomena. If facilitating such encounters is not in fisheries/wildlife management mandates now, this needs to change as we meet the challenges—and the opportunities—of the Anthropocene.

## 8. Conclusions

We have argued that, in cases where humans and wild animals come into overlapping and, at times, conflicting proximity, where various interests of both humans and wild animals need to be taken into account, philosophical accounts of animal ethics offer rich resources for adjudicating these cases in a more just manner than was seen in Freya’s case. A particular strength of our analysis is that one need not accept all three of the lenses—co-sovereignty, capabilities, and individuality—as compelling. As long as at least one is, a more just outcome can be secured in future cases.

## Data Availability

No new data were created or analyzed in this project. Data sharing is not applicable for this work.

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
