# Peer review of "The Life and Death of Freya the Walrus: Human and Wild Animal Interactions in the Anthropocene Era"

_animals, 2023, doi:10.3390/ani13172788_

Round 1

Reviewer 1 Report

The paper aims to establish guidelines for more just treatment of wildlife within human-wildlife conflicts or interactions, utilizing the euthanizing of Freya the walrus in Norway as the key case study. The authors provide a solid overview of competing claims, and outline three relatively recent philosophical frameworks that are relevant to the case. The case of Freya is a particularly significant and timely one, and the authors frame the case as an example of the kinds of conflicts that are likely to increase as anthropogenic climate change continues. 

Concerns arise in several areas. First, there are assumptions made about Freya's intentions and behaviors. The authors reference her desire for "socialization" at various points, but this is unclear from the descriptions. In fact, more description of Freya's behavior and connection to research on walruses more broadly would strengthen these claims. Second, there is a lack of clarity around the "we" that the authors often name as being responsible for or answerable to Freya or other animals as moral subjects with whom humans have particular kinds of relationships. Human-wildlife conflicts such as this one are perceived differently by different audiences (urban/rural, class-based, settler/Indigenous, etc.), and addressing the myriad ways that Freya was seen by particular actors in this case with more detail would only strengthen the paper overall. This is noted in the reviewer's comments at various points in the attached file. Third, the authors are claiming that the actions undertaken by the Directorate to euthanize Freya demonstrated a choice to prioritize property over the capabilities, rights, or life of Freya. However, there is not enough of a discussion of property and "harm" for the readers to get a sense of whether or not the authors think property can be violated or whether destruction of property is a harm at all. There is not consensus on this ethically, and so the authors should address this in some ways. Again, this is noted throughout the paper.  

This paper has strengths and could be a solid contribution to the journal. The authors note particular strengths at the end of their paper, that demonstrate that one need not accept all three of their principles in order to come to the conclusion that Freya was treated unjustly and to move toward fairer and more just treatment in future. As a reviewer, I agree with this, but I think this paper could be stronger in ways that will leave out some of the gaps in claims or information.

Reviewer 2 Report

Dear authors,

Thank you very much for sharing your manuscript. I was very interested to read a more detailed account of the Freya case which I only knew about from the media. I also appreciate your carefully laid-out argument.

The only weakness I grappled with is the perspective of care ethics. Here, I have the feeling that your argument gets anthropocentric, in so far as our duty to consider animals as individual subjects depends on us forming a relationship, and having some (emotional) benefit from it. What if Freya had not been perceived as a likeable person, what if our human affection is so tangled up with projections that our actions toward wild animals are not conducive to their physical and socio-cognitive well-being? What, on this level of analysis, would be objective criteria from the perspective of the animal? These questions about preserving animal subjectivity do not become visible in this text because killing Freya was such a drastic measure. However, I think this is a general shortcoming of care ethics, and you may be inspired to pick up on this critique a little bit.

Looking forward to seeing your text published,

kind regards

Reviewer 3 Report

This interesting article falls within the topic of animal welfare and their management in anthropic environments. It is a very well written and reasoned essay that also proposes an approach associated with the probable anthropogenic effect on current climate change scenario.

This essay makes a well-founded critique of the decision taken by the Norwegian Directorate of Fisheries to euthanize a walrus, named Freya by her international fan base, that repeatedly climbed onto docked boats at marinas throughout Europe in order to sunbathe and frolic, and that during its last month of life was essentially present in the Oslo Fjord, surrounded by onlookers, tourists and fans.

The authors list the events that occurred prior to the death of Freya, and propose a series of alternatives to the euthanasia that the Norwegian government finally authorized. Apart from the fact that these alternatives are more in line with measures adjusted to the Bern Convention on the protection of species, this essay put forward the fact that encounters such as this, between wild fauna and humans, particularly in marinas areas, may become more common under current climate change scenario, which may be causing wild animals to move away from their native distribution areas. The euthanasia of "nuisance" animals is a very bad precedent that must be avoided or considered as a truly last option in the relationship between human and non-human animals.

In order to organize their reasoning, authors pointed out four key competing interests in adjudicating a proper response to Freya’s arrival in the fjord, and outlined three distinct philosophical lenses to weigh such competing interests in a more just and complete way.  Furthermore, three background premises were considered throughout the manuscript: a) animals have the same right to use the planet for survival as humans; b) there is no clean division between ‘wild’ and ‘civilized’ space; and c) we humans have moral responsibility for anthropogenic climate change.

Consequently, I consider that this is a relevant and timely manuscript worth to be published.

Some minor comments/clarifications.

2.1. Timeline of events. As it is written in the manuscript, it may appear that Freya was only seen in the Dutch coast before reaching Norwegian marinas, while she actually was recorded in the coast of several countries, i.e. Sweden, Denmark, England, Germany, the Netherlands and the Shetland Islands since 2019 (https://www.google.com/maps/d/u/0/viewer?hl=no&ll=58.3950945852862%2C7.3484551969395975&z=5&mid=1dWCp2CObfw6_l-v378GIf2VD9SOjOF-4). Please add this information.

Lines 521-522. Is there any evidence that Freya fed on invasive oysters? Please clarify and, if so, provide more information on that (such as the oyster species in question).

Finally, I miss a short account on walrus behavior, their population dynamics and basic ecology, to put into context Freya’s behavior and requirements.

Reviewer 4 Report

The authors use the flagship case of Freya the Walrus to explore the causes and options for resolution of human-wildlife conflict in the Anthropocene. They argue that the causes were anthropogenic at global (climate change) and local level (economic impact on marina boat owners) and the resolution (euthanasia of Freya) was inappropriate. The scholarship is narrow and reliant on extracts from books by Nussbaum (2022) and Donaldson & Kymlicka (2021). They have not explored the extensive literature on compassionate or dispassionate approaches to conservation, especially where lethal interventions occur. Nor have they considered other high profile flagship cases like ‘Cecil the lion’ killed in Zimbabwe for a predominantly venal motive. The commentary is quite lengthy, and the conclusions well argued so I do not recommend any lengthy addition but suggest some recognition of a larger literature is appropriate. This is particularly true when they bring compassion into their arguments. As a commentary the manuscript serves a useful purpose of raising issues and solutions for a high profile and internationally significant human-wildlife conflict. It will likely provoke further interest and debate.

The manuscript is well-written I have some minor suggestions as follows:

Line 11: argument why

Line 22: just, going

Line 66: We conclude our

Line 152: euthanasia.

Lube 279: of argument for

Line 304: against particular

Line 320: “is also of a piece with” – incomprehensible?

Line 463: significant that Freya

Line 664-5: interests that were

See minor corrections suggested in above

Reviewer 5 Report

In my opinion, the work "The Life and Death of Freya the Walrus: Human and Wild Animal Interactions in the Anthropogenic Era" is an original work of interest for the publication The work is based mainly on the ethical and philosophical part of the relationship between man and the animal world and in general on the relationships between living organisms and not on our planet. From my experience as a researcher of large  carnivores and in particular of bears, in Italy I am aware how important it is to feel and consider not only technical management aspects and scientific research that can provide through the collection of data in a rigorous manner, but also the most philosophical and ethical aspect combined with the needs of the communities local. In Italy, for example, compared to Norway, we have a the population of almost 4,000 wolves without any control plan as well and despite tragic events that involved the local population in Trentino with bears, the majority of the population is convinced that in the  wild areas, however, transitional areas (e.g. areas for livestock  activities) should always be given priority to the needs of wild species, especially large carnivores; on the contrary in Norway life is allowed only 4-6 packs of wolves throughout the territory in a well-defined area where their presence is not to the detriment of hunting and breeding, as well a maximum of 27 reproductions of wolverine is tolerated... in the face of a research effort unimaginable in other European countries, whose main purpose is to provide information on whether it is necessary to intervene on species that have exceeded numerical limits or specific behaviors. Another case that reminds those of Freya and the case of Bruno the bear who left from Trentino came to Bavaria and enthusiastically welcomed and then in a short time or downcast because considered dangerous, even if there were other management solutions to be able to apply.  Having said this, in view of the nature of the work, it is difficult for me to assess the adequacy of the data collected or the scientific approach, which in fact does not exist as we understand it ,  as well as the philosophical  approach that I share based on three pillars,  but I can and must suggest to integrate the work in particular in paragraph 2.2 with scientific papers and references, that can support the thesis of climate change that forces species to use areas they once did not use, as this aspect is brought as a supporting element to justify approaches other than what the Norwegian authorities have done and in general as a key to changing the paradigm, all without distorting the nature of the article. The work shows a certain weakness in that the bibliography is also based on many press articles and should be implemented in the scientific part s well as the framing of nature management in Norway from the legislative point of view and also bring an overall view of how and why wild animals are managed in this country differently from other European countries, recalling that Norway is not a member of the EU and therefore does not apply the Habitats and Birds Directive.  In paragraph 2.1 I also suggest using a table with the milestones of the story. As it would be to implement Chapter 7 with more information on the systems of prevention and mitigation of the damages from fauna adopted in Norway. I think that even an image and more images and/ or diagrams on the logical framework and the pillars of this article should be considered as an element of simplification in reading as well as if possible some photos of Freya.

Round 2

Reviewer 5 Report

Thanks for the answers and partial integrations. Anyway, I’m still puzzled about the number and type of quotes supporting the importance of climate change as far as Freya’s behavior. I also don’t understand the first answer, which is what "empirical approach" means. and then the comment "it doesn't make sense to compare the empirical approach to the philosophical one",  when the article, as described by the authors, with the philosophic approach,  is based (but not only) on observations often empirical based on comments his newspapers (not scientific) or a few scientific quotes. In addition, the article wants to travel winding roads that come to recommendations for management and policies. I wonder if the authors are convinced that a political and managerial approach can be based only on a philosophical approach. without taking into account national policies and social and economic contexts and perceptions?

I remain of the idea that the article is very interesting but lacks some transversality (remember how much weight is given to climate change in this work) that seeks but then does not find

Author Response

Thank you so much for your very helpful comments. 

Please see the attached file for our responses!
